# Platelet and HIV Interactions and Their Contribution to Non-AIDS Comorbidities

**DOI:** 10.3390/biom13111608

**Published:** 2023-11-02

**Authors:** Thomas Awamura, Elizabeth S. Nakasone, Louie Mar Gangcuangco, Natalie T. Subia, Aeron-Justin Bali, Dominic C. Chow, Cecilia M. Shikuma, Juwon Park

**Affiliations:** 1Department of Tropical Medicine, Medical Microbiology, and Pharmacology, John A. Burns School Medicine, University of Hawai‘i at Mānoa, Honolulu, HI 96813, USA; awamura@hawaii.edu (T.A.); nsubia@hawaii.edu (N.T.S.); aganirbali@gmail.com (A.-J.B.); 2University of Hawai‘i Cancer Center, University of Hawai‘i at Mānoa, Honolulu, HI 96813, USA; enakasone@cc.hawaii.edu; 3Department of Medicine, John A. Burns School of Medicine, University of Hawai‘i at Mānoa, Honolulu, HI 96813, USA; dominicc@hawaii.edu; 4Hawai‘i Center for AIDS, John A. Burns School of Medicine, University of Hawai‘i at Mānoa, Honolulu, HI 96813, USA; louiemag@hawaii.edu (L.M.G.); shikuma@hawaii.edu (C.M.S.)

**Keywords:** platelets, HIV, viral persistence, inflammation, coagulation, HIV complications, antiretroviral therapy

## Abstract

Platelets are anucleate cytoplasmic cell fragments that circulate in the blood, where they are involved in regulating hemostasis. Beyond their normal physiologic role, platelets have emerged as versatile effectors of immune response. During an infection, cell surface receptors enable platelets to recognize viruses, resulting in their activation. Activated platelets release biologically active molecules that further trigger host immune responses to protect the body against infection. Their impact on the immune response is also associated with the recruitment of circulating leukocytes to the site of infection. They can also aggregate with leukocytes, including lymphocytes, monocytes, and neutrophils, to immobilize pathogens and prevent viral dissemination. Despite their host protective role, platelets have also been shown to be associated with various pathophysiological processes. In this review, we will summarize platelet and HIV interactions during infection. We will also highlight and discuss platelet and platelet-derived mediators, how they interact with immune cells, and the multifaceted responsibilities of platelets in HIV infection. Furthermore, we will give an overview of non-AIDS comorbidities linked to platelet dysfunction and the impact of antiretroviral therapy on platelet function.

## 1. Introduction

Since the discovery of the Human Immunodeficiency Virus (HIV) in 1983, it has been estimated that 85.6 million people have been infected with the virus. According to the United Nations Programme on HIV/AIDS (UNAIDS), there are approximately 39 million people living with HIV (PLWH) globally as of 2022. Antiretroviral therapy (ART), which suppresses viral replication, has dramatically reduced the morbidity and mortality arising from HIV-induced immunosuppression and allowed PLWH to attain a standard of living like that of uninfected individuals [1,2]. Although HIV has become a manageable chronic disease, PLWH are at a disproportionately higher risk of developing early-onset chronic comorbid conditions [3,4,5], contributing to reduced health outcomes, increased complexity of medical management, and burden on healthcare systems [6,7,8]. This reality highlights the critical need for defining the biological mechanisms that drive the accelerated development of non-AIDS comorbidities to develop strategies to mitigate these processes [9,10].

HIV-1 primarily interacts with and infects T cells, macrophages, and dendritic cells. Significant progress has been made in understanding viral entry into these cells and the functional consequences of this process [11,12,13]. However, HIV-1 has also been observed to interact with other immune cells and their products [14,15,16]. Far less is known about how these interactions occur and their role in the pathogenesis of comorbid conditions. Thus, a better understanding of these host–pathogen interactions is critical for providing new opportunities to develop improved HIV-directed therapies, as well as better guidelines for screening and managing non-AIDS comorbidities. While platelets play an essential role in normal hemostasis [17], a growing body of evidence suggests that platelets play opposing roles in chronic HIV infection and contribute to the development of comorbid conditions [18,19,20]. Current knowledge suggests they can serve both as a host defense against infection, activating both innate and adaptive immunity, while simultaneously acting as a transient viral reservoir to promote the persistence of infection and viral spread [18,21]. Further, they are also involved in regulating chronic inflammation and endothelial activation [22,23], which can lead to the development of comorbid conditions. In this review, we discuss the interactions between platelets and HIV, focusing on the consequences of platelet–immune cell interactions and the clinical manifestations that arise from platelet dysregulation in HIV infection, as well as the impact of ART on platelet function.

## 2. Role of Platelets in HIV Infection

### 2.1. Platelet and HIV Interaction

HIV-1 internalization by mature megakaryocytes and platelets was first observed by ultrastructural analysis following co-culture of an HIV-infected T lymphocyte line with human platelets and marrow-derived megakaryocytes in 1990 [24]. Subsequent studies have demonstrated that internalization of HIV is dependent on platelet activation, which leads to endocytosis of viral particles and association with open canalicular systems [16,25].

Platelets sense HIV-1 through several receptors that allow platelets to interact with HIV-1 and induce their activation, as shown in Figure 1. These include CXC chemokine receptor (CXCR)1, 2, and 3 [26,27], CC chemokine receptor (CCR)1, 3, and 4 [27,28], Dendritic Cell-Specific ICAM3-Grabbing Non-integrin (DC-SIGN) [29], and C-type lectin receptor 2 (CLEC-2) [30,31]. In addition, viral proteins that are released from HIV-1 infected cells, such as the trans-activator of transcription (Tat), can also bind to integrin β_3_ and CCR3 on platelets and activate them [32,33].

Platelets respond to damage-associated molecular patterns (DAMPs) released by infected cells and pathogen-associated molecular patterns (PAMPs) produced by HIV via various pattern recognition receptors (PRRs), such as the C-type lectin receptors (CLRs) DC-SIGN and CLEC-2 [19,34,35,36]. These CLRs recognize mannose-terminal-containing pathogen-associated carbohydrates and bind to the glycosylated domains of the glycoprotein 120 (GP120) subunit of the HIV envelope (E) protein [19,30]. Once bound, the virus can be internalized via endocytosis, where it can either remain sheltered within endocytic vesicles or exposed to secretory products via interactions with α-granules [16,21]. α-granule-derived proteins, including cytokines and chemokines such as platelet factor 4 (PF4), interleukin (IL)-1α, 1β, 4, 6, 8, 10, 13, tumor necrosis factor (TNF)-α, interferon (IFN)-α, β, and γ can directly kill HIV or inhibit viral replication (Figure 1) [37,38,39,40,41,42,43,44].

### 2.2. Role of Platelets in Modulating Innate and Adaptive Immune Clearance of HIV

Platelets can directly inhibit HIV replication, which can lead to viral clearance [16,37,45]. However, platelets themselves are insufficient to mediate the complete clearance of viral infection. Instead, they primarily serve to aid in the recruitment of immune cells and modulation of inflammatory response [18]. This allows platelets to serve as an intersection between innate and adaptive immune responses to HIV, allowing for a more rapid response to the virus. Platelets interact with various types of innate immune cells, particularly monocytes, macrophages, and neutrophils, by inducing the release of platelet-derived cytokines and chemokines, thus exerting a protective effect in the physiologic response to disease and control of viral infection [21,46]. These secreted proteins not only increase the capacities of phagocytes to serve as effectors and allow for leukocyte recruitment, but they can also induce anti-viral pathways in neighboring cells, making them less susceptible to viral infections [21,46].

Platelet activation occurs when platelets are exposed to DAMPs or PAMPs, promoting various coagulation and immune pathways, resulting in drastic alterations to the phenotype of the platelets [47]. This phenotypic shift promotes platelet coagulation, adherence, and production and release of granule-stored compounds that can activate nearby leukocytes or have antimicrobial properties [47]. Activated platelets upregulate the expression of P-selectin, allowing them to both bind to endothelial cells lining the vasculature and adhere to circulating leukocytes (Figure 1) [48]. Platelets infected by HIV or activated by Tat also secrete large amounts of CD40 ligand (CD40L) to recruit leukocytes, including monocytes, neutrophils, and T-cells, and form immune complexes [49].

HIV treatment naïve individuals have increased levels of platelet-monocyte aggregates (PMA), which causes mutual activation of these cell types, specifically with CD16+ monocytes (non-classical and intermediate monocytes) and platelets [50,51]. Higher levels of circulating PMA are correlated with viremia and increased sCD163 levels in HIV-1 infection [50]. Platelets can form complexes with neutrophils through a variety of mechanisms, including binding between platelet Ibα (GP1bα) and neutrophil Mac-1, platelet P-selectin and neutrophil P-selectin Glycoprotein Ligand-1 (PSGL-1), or platelet integrin α_II_β_3_ and neutrophil Mac-1 via soluble fibrinogen [52]. These platelet–neutrophil interactions result in neutrophil extracellular traps (NETs) formation, in which neutrophil activation leads to the expulsion of DNA and enzymes that trap and destroy pathogens, thereby preventing viral amplification and dissemination (Figure 1) [21,53].

Platelet activation, leading to the expression of surface lectins, Fc-receptors, and storage of complement proteins such as C3, C4a, and complement factor H produced in α-granules can result in the activation of all three pathways of the complement system, as well as the perpetuation of complement activation [54]. While the Fc-receptors expressed on neutrophils are relatively low affinity as they cannot bind IgG monomers directly and only antigen-antibody complexes [55], they nonetheless can lead to activation of classical complement pathway when platelets bind to antibody-bound virions or infected cells, leading to virolysis [56]. Activation of toll-like receptor (TLR7) by single-stranded RNA viruses such as HIV and influenza has led to the significant release of C3 from platelet granules, aiding in the formation of NETs [11]. Furthermore, complement opsonization of both virus particles and infected cells also promotes viral clearance by phagocytic cells, in addition to releasing large amounts of C3a and other complement components via complement convertases formed at the surface of opsonized targets that serve as chemoattractants for nearby immune cells (Figure 1) [56].

The role of platelets in modulating adaptive immune function in HIV infection is poorly understood. Activated platelets can serve to protect CD4^+^ T cells from HIV infection when co-cultured in-vitro, indicating that platelets may serve as a potent first-line defense against infection of susceptible cell types [45]. This may be due to the significant release of CXCL4 in response to HIV by platelet α-granules during platelet activation, which inhibits viral entry into nearby T-cells [45,57]. Platelets can also transfer HIV antigens or intact virions to APCs to initiate humoral immune responses [58]. However, it is still largely unknown whether this mechanism significantly impacts establishing antibody response against HIV.

### 2.3. Role of Platelets in Sheltering HIV and Viral Persistence

While platelets can assist in the initial suppression and clearance of HIV, they can also contribute to viral persistence by potentially serving as a viral reservoir (Figure 2) [59]. In fact, the internalization of viral particles by platelets is not unique to HIV and has been shown in influenza, respiratory syncytial virus (RSV), and Dengue virus (DENV) [60,61,62]. Furthermore, the hepatitis C virus (HCV) has been shown to bind to platelet glycoproteins in order to hide from immune responses, aiding in the establishment and persistence of infection [63].

Several mechanisms by which platelets can promote HIV dissemination have been identified, which are illustrated in Figure 2 [64,65]. HIV virions internalized by platelets can remain viable within an open canalicular system (OCS) and endocytic vesicles not destined for destruction by association with α-granules Figure 2(A) [65]. These virions are shielded from HIV-neutralizing mechanisms, including antibodies, complement proteins, and activation of cytolytic effector cells. While HIV does not establish latency within platelets, they have been observed to interact with permissive cells such as CD4^+^ T cells, leading to direct viral transmission to uninfected cells Figure 2(B) [25]. Infected, activated platelets can also produce microparticles, membrane vesicles, that may contain HIV virions and/or express CXCR4, allowing for HIV transfer and/or the coreceptor to CXCR4-negative cells (Figure 2) [38]. Competent virions may survive within platelet aggregates, promoting infection of phagocytic cells as these cells attempt to clear thromboses from the vasculature and tissue Figure 2(C) [25,64]. With platelet activation, it also increases soluble factors like CD40L and thrombin receptor activator peptide 6 (TRAP6) to increase platelet-virus interactions Figure 2(D) [25]. Therefore, the activation of platelets by Tat may be a cause of endothelial dysfunction, enhancing the recruitment of virus-susceptible leukocytes [66,67].

Platelets thus appear to serve as transient transporters of HIV [64] and, due to their widespread distribution throughout the body, are believed to lead to viral dissemination [65]. Indeed, platelets have even been suggested to play a role in central nervous system (CNS) spread, as activated platelets can cross the blood-brain barrier (BBB). However, the exact mechanism by which this occurs has yet to be shown [68]. Interruption in ART may result in the re-emergence of HIV from viral reservoirs within platelets [69].

### 2.4. Role of Platelet Activation in the Pathophysiology of HIV

During HIV-1 infection, an exaggerated systemic inflammatory response leads to platelet activation and dysfunction, promoting a hypercoagulable state and increasing the risk for thrombosis [70,71]. Several mechanisms driving platelet activation in the context of HIV have been identified, including the recognition of PAMPs and DAMPs, such as Tat, by PRRs and CLRs on platelets [21,36]. HIV trafficking from early to late endosomes leads to platelet activation by binding of TLRs and cytosolic nod-like receptors (NLRs) [36]. HIV-1 E protein increases circulating inflammatory cytokines, including IL-1α, IL-1β, IL-2, IL-6, TNF-α, IFN-α/β, NFkB, and IFN-γ, and TNF-α promoting oxidative damage, which leads to endothelial dysfunction and triggers platelet activation [21,35]. Activation of platelets with subsequent generation of platelet microparticles leads to upregulation of tissue factor expression [72,73], activating the coagulation cascade and downstream production of thrombin and fibrinogen [74].

Platelet activation leads to upregulation of CD40L expression in α-granules and translocation to the cell surface [75]. Cleavage of platelet surface CD40L occurs rapidly and leads to the production of soluble CD40L (sCD40L), which is believed to play an essential role in platelet activation and pathogenesis of atherosclerotic plaques and thromboses [76,77,78]. sCD40L binding to platelet α_IIb_β_3_ stabilizes arterial thrombi by upregulating P-selectin expression [79]. CD40L is also expressed in many adaptive and innate immunity cells, promoting the formation of platelet-leukocyte aggregates that contribute to atherosclerotic thrombus formation [80]. For example, platelet P-selectin binds to PSGL-1 on monocytes, facilitating complex formations between platelets and monocytes [19,21,81] and promoting the secretion of proinflammatory cytokines and tissue factors [82,83,84]. Furthermore, endothelial activation leads to dysregulated von Willebrand Factor (vWF) production, which promotes platelet aggregation and adhesion to exposed dysfunctional endothelium [85] and contributes to atherosclerotic thrombus formation. A recent study found that elevated vWF antigen levels correlate with ischemic stroke among PLWH on ART [86]. Overall, HIV infection sensitizes platelets to procoagulant and proinflammatory factors and induces monocytes to do the same, creating a positive feedback loop underlying chronic inflammation and thromboembolic risk complications [81].

## 3. HIV Complications Due to Platelet Dysfunction

### 3.1. Thrombocytopenia

Thrombocytopenia is defined as a platelet count of less than 150,000/µL. It has been reported in 5–30% of PLWH [87,88,89,90,91,92,93] and may be one of the first indicators of infection [91], although it is observed across all stages of HIV infection [94]. In healthy adults, morbidity due to thrombocytopenia is related to potential hemorrhage, with a high risk for spontaneous bleeding typically observed at a platelet count of less than 10,000/µL [95]. In PLWH, the risk of bleeding is observed at higher platelet counts. It is often associated with co-existing comorbidities, including chronic liver disease leading to portal hypertension and increased rates of variceal bleeding [96] and hemophilia [97]. The development of thrombocytopenia in PLWH is multifactorial, arising from ineffective platelet production and immune-mediated destruction.

Hypoproliferative thrombocytopenia (thrombocytopenia due to platelet underproduction) arises from impaired growth factor production and ineffective megakaryopoiesis. Comorbid conditions, such as co-existing viral infections with hepatitis B [98] and C [99], as well as other etiologies that lead to cirrhosis, result in impaired thrombopoietin production [100] and portal hypertension with resulting hypersplenism and platelet sequestration [101]. Direct viral infection of megakaryocyte progenitors leads to impaired megakaryocytic differentiation [102,103], and infection of megakaryocytes leads to apoptosis [104].

Immune activation also results in decreased survival of platelets in circulation [105]. Reticuloendothelial activation enhances platelet destruction in part due to direct platelet activation. CXCR4-mediated endocytosis of viral particles, with subsequent exposure of virions to alpha granule contents, leads to the activation and expression of P-selectin and platelet clearance by circulating macrophages [16].

Even more common than reticuloendothelial activation is secondary immune thrombocytopenia (ITP). HIV-associated ITP was first reported in 1982 [106] and has since been documented as occurring in 6–15% of PLWH [107]. During HIV infection, immune activation results in the production of immunoglobulins that recognize or cross-react with platelet glycoproteins [108,109] and activate complement, leading to the formation of circulating immune complexes consisting of immunoglobulins, complement proteins, and platelets [110,111]. Indeed, immunoglobulins that recognize a common epitope between HIV p24 or GP120 and platelets [112,113,114], as well as those that recognize the platelet glycoprotein GPIb/IX and GP IIb/IIIa [115,116,117,118,119], have all been detected in the circulation of PLWH. These antibodies can be detected in HIV-seropositive patients even before the onset of symptoms or thrombocytopenia [120], suggesting that immune dysregulation develops in the earliest phases of HIV infection. Subsequent splenic sequestration of these immune complexes removes platelets from circulation [121].

The severity of thrombocytopenia is directly correlated with HIV viral load, and initiation of ART has been shown to improve platelet counts in patients with adequate viral suppression [122,123,124], although normalization of platelet counts may not occur [125]. Despite appropriate optimization of ART, persistent immune-mediated thrombocytopenia can be managed with standard therapies used for patients with primary ITP, including corticosteroids, intravenous immune globulin, and Rh_o_(D) immune globulin. Rituximab may also be utilized, although responses tend to be short-lived, particularly in those with co-existing hepatitis C viral infection [126,127,128]. Splenectomy has also been observed to be effective in improving platelet counts in those refractory to medical treatment [129,130,131,132,133,134], although splenectomy is infrequently used due to complications related to thrombosis and infection [135]. More recently, case reports and series have demonstrated the efficacy of thrombopoietin receptor agonists in patients with refractory HIV-associated ITP [136,137,138].

### 3.2. Thrombotic Microangiopathies

Thrombotic microangiopathies (TMAs) are a spectrum of rare but potentially life-threatening syndromes characterized by endothelial damage. This triggers platelet activation and aggregation, leading to platelet consumption and thrombocytopenia, microthrombi formation, which results in microvascular ischemia and end-organ dysfunction, and the mechanical destruction of red cells as they pass through partially occluded microvasculature which gives rise to hemolytic anemia [139].

The association between HIV infection and thrombotic microangiopathies was first reported in the 1980s [140,141,142]. The most commonly observed TMA manifestations in patients with HIV infection are secondary thrombocytopenic purpura (TTP) and secondary hemolytic uremic syndrome. Prior to the advent of ART, the incidence of a TMA was estimated to be approximately 1.4% [143] and has since improved to approximately 0.3% [144]. HIV-associated TMAs are more frequently observed in patients with higher viral loads and more advanced stages of disease [145,146]. However, as with HIV-associated ITP, HIV-associated TTP/HUS can be observed in both patients who are ART-naïve and patients on ART, including those who demonstrate complete virologic suppression [147]. Indeed, despite the availability of anti-retroviral therapy in South Africa, morbidity and mortality due to HIV-associated TTP remains very high, with estimates suggesting an incidence 15–40 times greater than that observed in non-infected individuals [148].

TTP/HUS can be differentiated based on presenting symptoms. While thrombocytopenia, microangiopathic hemolytic anemia, and acute kidney injury are associated with both syndromes, TTP includes the additional features of fever and neurologic dysfunction [149]. TTP and HUS can be further distinguished based on the triggering event that leads to endothelial damage.

Acquired TTP results from the severe, antibody-mediated deficiency of a-disintegrin-and-metalloproteinase-with-thrombospondin-motifs 13 (ADAMTS-13) and enzyme activity less than 5% of normal diagnostic [150]. ADAMTS-13 is required to cleave ultra-large vWF multimers, and enzyme deficiency leads to the deposition of ultra-large vWF multimers in endothelial surfaces and subsequent platelet activation [151]. While HIV-associated TTP is frequently associated with ADAMTS-13 deficiency, patients may not present with severe deficiency, and inhibitory autoantibodies are not always detected [152,153,154], suggesting that chronic inflammation and endothelial activation contribute to the pathogenesis of TTP. Interestingly, severe enzyme deficiency appears to be associated with higher CD4 counts and fewer HIV-related complications [155]. In contrast, patients with HIV-associated TTP who demonstrate great enzyme activity are observed to have higher rates of TTP-associated death as compared to non-infected patients with similar characteristics [155].

Secondary HUS is characterized by uncontrolled complement activation via the alternative pathway. Persistent complement activation leads to endothelial damage and activation, which ultimately results in a similar pattern of platelet activation and aggregation, microthrombi formation with associated microvascular ischemia, and hemolytic anemia [156].

Treatment for HIV-associated TMAs is similar to that of other forms of acquired TMAs. It involves supportive care with plasma transfusion until therapeutic plasma exchange and immunosuppressive therapies (e.g., steroids, rituximab) can be initiated [145,146,152]. Durable remissions can be obtained with the optimization of ART [145,152,157], and developing a TMA while on ART could suggest evolving therapeutic resistance [145]. At present, it is unclear whether other approved therapies for TTP would also be effective for managing HIV-associated TMA. However, case reports suggest that complement inhibitors like eculizumab may be effective [158,159].

Acute disseminated intravascular coagulation (DIC) is another form of TMA arising from overt immune activation and endothelial dysfunction. This activates the coagulation cascade, the fibrinolytic system, and platelets. While microvascular thromboses are a feature of DIC due to the activation of procoagulants and platelets, this is a consumptive process that ultimately results in a bleeding diathesis [160]. While acute DIC is observed in PLWH, it is rarely observed in earlier stages of infection HIV infection and is typically attributed to etiologies other than HIV infection itself, most commonly disseminated infection [161]. Management of acute DIC involves supportive care, appropriate management of the precipitating etiology, and ART initiation.

### 3.3. Venous Thromboembolism

Venous thromboembolism (VTE) is the formation of blood clots in the deep veins of the extremities and pelvis. It is associated with pulmonary embolism, in which blood clots travel proximally from distal veins to the pulmonary arteries [162]. In PLWH, the annual incidence of VTE has been reported to range from 0.19 to 7.63% depending on the series, and the risk of thrombosis is 2–10 fold greater than that that in non-infected patients (reviewed by Zhang et al. [163]).

The predominant contributing factor to this increased risk of thrombosis is chronic inflammation, which leads to a generalized hypercoagulable state. Chronic, compensated DIC leads to a consumptive coagulopathy that results in deficiencies of multiple anticoagulants, including proteins C and S, heparin cofactor II, and antithrombin [164]; in patients who have had immune reconstitution with ART, autoimmunity is frequently observed, resulting in the production of prothrombotic auto-antibodies such as antiphospholipid antibodies and lupus anticoagulant. In patients with later stages of HIV infection, opportunistic infections (particularly cytomegalovirus, *Mycobacterium tuberculosis*, Mycobacterium avium complex, and *Pneumocystis*) and HIV-associated malignancies also contribute to this hypercoagulable state [163]. Further, side effects of medications, including protease inhibitors used to suppress viral replication and the appetite stimulant megestrol (a synthetic progestin), also promote a hypercoagulable state [163].

Venous thromboses predominantly comprise fibrin and erythrocytes, with few platelets [165]; thus, platelets have been generally thought to play a limited role in thrombus formation. Commonly recognized predisposing factors that lead to venous clot formation are low flow states, endothelial damage, and hypercoagulability (Virchow’s triad). Low flow states that lead to local hypoxia and the generation of reactive oxygen species, mechanical injury, or functional dysfunction resulting from systemic inflammatory states (e.g., sepsis) disrupt the normal anticoagulant properties of venous endothelium promoting expression of endothelial adhesion proteins, including P-selectin and platelet activation [166]. Leukocytes adhere to P-selectin via PSGL-1, where they liberate TF, activating the coagulation cascade and initiating thrombus formation [167].

However, in PLWH, chronic platelet activation is believed to play a more active role in the pathogenesis of VTE compared to the general population. Activated platelets can directly mediate the trafficking of immune cells to sites of endothelial damage and are also associated with increased expression of tissue factor [72]. Additionally, activation of platelets leads to the P-selectin-mediated formation of platelet-activated T cell complexes [35], which can result in the recruitment of effector/memory T cells to sites of injury, further promoting local inflammation, endothelial activation, and hypercoagulability [168]. PLWH can also demonstrate thrombocytosis, which may be more pronounced after ART initiation [169] or may be secondary to chronic inflammation resulting in elevated levels of IL-6, which promotes thrombopoiesis [170].

Anticoagulation recommendations, including duration of therapy, for VTE in PLWH are generally similar to those for non-infected patients [171]. The most commonly used anticoagulants in PLWH are low molecular weight heparin and vitamin K antagonists, mainly due to interactions with ART. While warfarin does interact with ART, patients on warfarin can be closely monitored in the outpatient setting to ensure that appropriate therapeutic levels are maintained [172].

As PLWH were excluded from the initial studies that evaluated these direct Xa inhibitors for the management of VTE [173,174,175,176], there is limited data on these agents’ use in PLWH. Currently, available data suggest that perhaps dabigatran, a direct thrombin inhibitor, maybe the safest alternative to warfarin at present [177]. Although etexilate, the prodrug of dabigatran, is a PgP glycoprotein, which can be inhibited by protease inhibitors and lead to increased bioavailability of dabigatran [178], this interaction can be mitigated by administering dabigatran two hours prior to the protease inhibitor [178]. However, this does require careful patient instruction and adherence to this dosing schedule. With respect to direct factor Xa inhibitors, ART can alter the activity of cytochrome P450 3A4, which may lead to subtherapeutic or supratherapeutic levels [178]. As PLWH on ART were excluded from trials evaluating direct factor Xa inhibitors, little data are available on safety and efficacy in these populations. Local experience and limited data sets and case series [179] suggest the safety and efficacy of direct Xa inhibitors; however, further evaluation is needed in formalized clinical trials.

### 3.4. Cardiovascular Disease

In PLWH, arterial thrombosis is observed predominantly in the form of cardiovascular disease. Cardiovascular disease (CVD) arises from atherosclerotic narrowing of arteries with or without plaque rupture and thrombus formation [180]. The most common manifestation of CVD in PLWH includes coronary artery disease, for which patients are at a 50% increased risk of acute myocardial infarction [181]; stroke, for which PWLH are at a 20–60% risk [182,183,184]; and peripheral arterial disease, for which PLWH are at a 20% increased risk [185]. The high prevalence of metabolic syndrome, which includes hypertension, dyslipidemia, diabetes mellitus, obesity, and specific behavioral patterns, particularly tobacco use, all contribute to an increased risk of CVD in PLWH [186]. Furthermore, the development of metabolic syndrome in PLWH is associated with chronic inflammation and immune activation, which persist despite viral suppression by ART [187].

Chronic inflammation leads to endothelial dysfunction and activation of atherogenic macrophages, resulting in the formation of atherosclerotic plaques [188] that are less likely to be calcified [189] and more prone to plaque rupture [190]. Plaque rupture provides the nidus for thrombus formation, while platelet activation and aggregation play a critical role in arterial thrombosis [191]. In metabolic syndrome, circulating low-density lipoproteins activate platelets, increasing oxidative stress and promoting resistance to the platelet-inhibitory effect of nitric oxide [192]. As is the case for the development of venous thrombosis, upregulation of P-selectin and tissue factor expression recruit leukocytes to promote local endothelial damage and inflammation and activate the coagulation cascade [72].

Several studies have sought to identify the optimal prophylactic therapy for preventing future cardiovascular risk in patients with HIV. While the United States Preventative Services provides detailed guidelines regarding the use of antiplatelet therapy for primary prevention in uninfected patients with high CVD risk, there are limited data to support the development of similar guidelines for the use of antiplatelet therapy for primary prevention in PLWH [193]. The indications and contraindications of the use of antiplatelet agents (such as aspirin, clopidogrel, or prasugrel) in HIV are similar to those of the general population.

Indeed, variable results have been obtained regarding the efficacy of aspirin in inhibiting platelet function and suppressing the production of inflammatory markers [194,195]. In contrast, clopidogrel, a platelet P2Y_12_ receptor antagonist, appeared to provide superior inhibition of platelet activity as compared to aspirin and was associated with a significant reduction in systemic inflammatory markers, suggesting that clopidogrel may be beneficial as a prophylactic medication to reduce the risk of cardiovascular events [196]. However, whether this is associated with a reduction in CVD outcomes remains unknown.

The most recent American Heart Association (AHA) and American College of Cardiology (ACC) recommendations for cardiovascular risk reduction in PLWH were provided in 2019 [197]. Current recommendations include a focus on lifestyle modification (including tobacco cessation) and use of statins (as with anticoagulants, these need to be carefully selected to prevent interactions with ART that may cause supratherapeutic levels), without a recommendation for primary prophylaxis with antiplatelet therapy or systemic anticoagulation [197]. While professional societies like the AHA and ACC provide guidance for primary CVD risk reduction in PLWH, these recommendations are not the standard of care. Indeed, statin therapy for primary prevention remains a controversial topic. However, there is a growing body of evidence supporting that statin therapy may reduce the risk of the first CVD event in PLWH. Most recently, the REPRIEVE trial, which evaluated 4 mg/day pitavastatin vs. placebo in PLWH with low to moderate traditional CVD risk on stable ART, demonstrated that daily pitavastatin is associated with a significantly reduced risk of major CVD events (4.81 per 1000 person-years compared to 7.32 per 1000 person-years in the placebo control group, hazard ratio, 0.65; 95% confidence interval [CI], 0.48 to 0.90; *p* = 0.002) during a median follow up of 5.1 years [198]. These and similar studies support the notion that statins for PLWH are potentially beneficial in reducing the risk of adverse CVD events. After a first episode of acute coronary syndrome, PLWH are at increased risk of recurrent ischemic events and have poorer outcomes as compared to non-infected patients [199,200], likely related to both biologic factors and non-biologic factors [197]. Interventions, including percutaneous coronary intervention, coronary artery bypass grafting, and endovascular treatment for stroke, appear to be at least equally effective in PLWH as compared to non-infected patients. In contrast, several studies have demonstrated that dual antiplatelet therapy does reduce platelet activity but does not necessarily alter the risk for recurrent events [201]. In general, recommendations for secondary prevention include the application of standard therapeutic guidelines for the treatment of CVD, as well as consideration for additional work up and intervention based on contributing pathologies (e.g., vasculitis, immune reconstitution inflammatory syndrome, etc.) [197].

## 4. Effects of ART on Platelet Count and Activity in HIV-1 Infection

### 4.1. Effect of ART Initiation on Platelets in HIV-1 Infection

Thrombocytopenia is seen more commonly among PLWH not receiving ART with advanced disease. Among people with AIDS published in 1982 (pre-ART era), thrombocytopenia was noted in 40% of the patients [106]. The rates of thrombocytopenia have decreased in the era of cART A systematic review that evaluated participants from the Collaborations in HIV Outcomes Research/US (CHORUS) cohort (1997 to 2006), and participants from HIV Clinical Trials (1996 to 2004) found that the prevalence of thrombocytopenia was approximately 14% [202]). A study from the British Columbia Centre for Excellence in HIV/AIDS published in 2012 reported that 26% had low platelet count (<100 × 10^9^/L) [127]. In a series of ART-naïve PLWH from South Korea (N = 621) studied from 2005 to 2010, thrombocytopenia was found in 2.4% [203].

As described in previous sections, HIV can cause thrombocytopenia through various mechanisms [204]. Initiation of effective ART, leading to virological control and undetectable plasma HIV RNA, has therefore been found to lead to platelet level recovery among PLWH [202]. However, the patient’s nadir (lowest) platelet count serves as a predictor of platelet count recovery. For example, in the CHORUS Cohort, 23% of patients with severe thrombocytopenia never achieved a higher platelet count on follow-up [202]. It is estimated that thrombocytopenia arising from HIV-associated ITP or due to a direct effect of HIV on platelet production and survival resolves within one to five months from initiating ART [202].

### 4.2. Adverse Effects of ART on Platelets

Virological control through ART can lead to hematological recovery of HIV-associated immune-mediated cytopenias. However, specific antiretroviral agents have been associated with decreased platelet count (Table 1), although causal relationships between specific medications and platelet counts currently cannot be determined. HIV treatment typically involves a combination of two or three antiretroviral agents and post-marketing experiences are reported voluntarily from a population of uncertain size.

## 5. Conclusions

In conclusion, platelets are versatile cellular products now recognized to have multiple functional roles beyond their contributions to hemostasis. In HIV-1 infection, platelets can both promote and prevent viral eradication. Recent findings suggest that platelet infection and processing of viral particles lead to the interaction and activation of innate and adaptive immune cells, promoting viral clearance. However, unprocessed particles can remain within platelets and act as a transient reservoir for HIV-1, promoting viral dissemination. Direct infection of platelets promotes viral clearance, leading to low platelet counts in ART-treatment naïve PLWH. While most PLWH receiving ART demonstrate improvement in platelet counts to their baseline counts, some patients do not ever demonstrate platelet count recovery. HIV infection can lead to generalized inflammation, leading to autoimmune processes that alter platelet survival and function and platelet activation, thereby promoting chronic inflammation and endothelial activation to promote hypercoagulability and thrombosis. As platelets play a significant role in the pathogenesis of thrombotic complications in PLWH, particularly cardiovascular disease, therapies that specifically inhibit platelet activation may be a valuable strategy for reducing the risk of developing comorbid conditions. However, at present, there are limited data on the utility of antiplatelet therapy as a preventive agent in PLWH, highlighting the need for further studies.

## Figures and Tables

**Figure 1 biomolecules-13-01608-f001:**
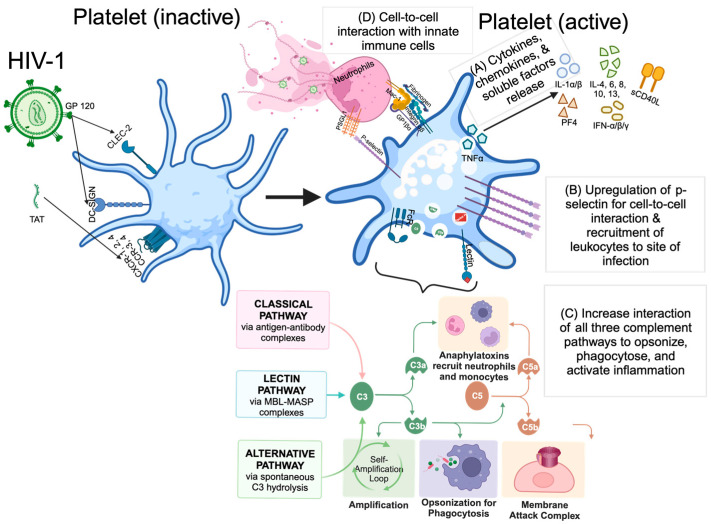
Downstream effects of HIV-mediated platelet activation: (A) Cytokine, chemokine, and soluble factors release, such as IL-1α/β, 4, 6, 8, 10, and 13. (B) Up-regulation of P-selectin for cell–cell interaction and recruitment of leukocytes to sites of infection, as well as to dysregulated endothelium. (C) Activation of all three complement pathways (classical, lectin, and alternative) to opsonize, phagocytose, and activate inflammation. (D) Interaction with innate immune cells (monocyte, macrophage, and neutrophils).

**Figure 2 biomolecules-13-01608-f002:**
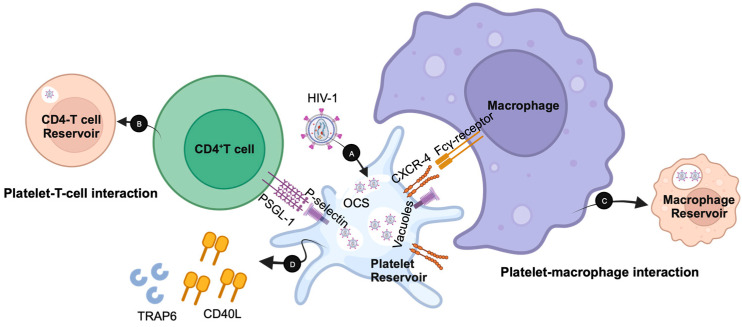
HIV-1 persistence after the initial activation of platelets (A) in becoming a transient reservoir in harboring HIV-1 in vacuole compartments and OCS, allowing the spread through CD4-T cell interaction (B), phagocytosis by macrophages (C), and increasing soluble factors (TRAP6 and CD40L) to increase HIV-1-platelet interaction (D).

**Table 1 biomolecules-13-01608-t001:** Summary of HIV antiretroviral therapy and rates of thrombocytopenia.

Drug	Rate of Thrombocytopenia *
**Nucleoside/nucleotide reverse transcriptase inhibitor (NRTI)**	
Abacavir	Grades 3/4: 1%
Didanosine	<1%, post-marketing and/or case reports
Emtricitabine	None reported in the FDA product label
Lamivudine	Adults: 4%; Children: 1%
Stavudine	<1%, post-marketing and/or case reports
Tenofovir disoproxil fumarate	None reported in the FDA product label
Tenofovir alafenamide	None reported in the FDA product label
Zidovudine	Infants, children, and adolescents, grades 3/4: 1%
**Non-nucleoside reverse transcriptase inhibitor (NNRTI)**	
Doravirine	None reported in the FDA product label
Efavirenz	None reported in the FDA product label
Etravirine	None reported in the FDA product label
Nevirapine	Thrombocytopenia rates similar to placebo
Rilpivirine	None reported in the FDA product label
**Protease inhibitor**	
Atazanavir	Grades ¾: 2%
Darunavir	None reported in the FDA product label
Fosamprenavir calcium	None reported in the FDA product label
Lopinavir/ritonavir	Grade 3/4: children: 4%
**Integrase inhibitor**	
Bictegravir	None reported in the FDA product label
Cabotegravir	None reported in the FDA product label
Dolutegravir	None reported in the FDA product label
Raltegravir	Post-marketing reports of thrombocytopenia
**Other pharmacologic category**	
Cobicistat **	None reported in the FDA product label
Enfuvirtide	<1%, post-marketing, and/or case reports
Fostemsavir	None reported in the FDA product label
Ibalizumab	Decreased platelet count < 50,000/mm^3^: 3%
Lenacapavir	None reported in the FDA product label
Maraviroc	None reported in the FDA product label

* Derived from the U.S. Food and Drug Administration Online Label Repository (available at https://labels.fda.gov/, accessed 20 September 2023) unless otherwise specified. ** Cobicistat is not an antiretroviral but a cytochrome P-450 inhibitor used as a pharmacokinetic enhancer for other antiretroviral drugs. Abbreviation: FDA, Food and Drug Administration.

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
