# Peer review of "Platelet and HIV Interactions and Their Contribution to Non-AIDS Comorbidities"

_biomolecules, 2023, doi:10.3390/biom13111608_

Round 1

Reviewer 1 Report

Comments and Suggestions for Authors

This review  summarizes platelet and HIV interactions during infection. The authors also discuss the role of platelet-derived mediators and their effects on immune cells in this process. Finally. The role of non-AIDS comorbidities linked to platelet dysfunction and impact of antiretroviral therapy is discussed.

Overall, this is an important review which should be of interest to both HIV and platelet scientists, as well as the general immune/inflammation community. This referee suggests a few points which should be considered by the authors .

 1)    The title of the review is very general, but the review deals primarily with platelet-HIV interactions. This should be reflected also in the title.

2)    The recognition that platelets can inhibit HIV replication, but also contribute zo viral persistence is very important, also clinically. This information is well described with many references, but a little bit difficult to digest. Could the authors prepare here a short table/figure to summarize the key facts ?

3)    The authors briefly touched the issue of antiplatelets. There are also many other drugs used for inflammatory diseases, which directly or indirectly affect platelets Are there conditions when antiplatelet drugs should be avoided with PLHW?

4)    While this review focusses on platelet HIV-interactions, it might help the reader if the authors have a very short paragraph comparing the HIV interactions with those of other viruses.      

Author Response

We would like to thank the Reviewer for their valuable time in reviewing and providing feedback to enhance our manuscript. We have carefully considered the feedback and addressed each concern within the newly submitted manuscript. Below, find included a point-by-point response to each comment.

1. The title of the review is very general, but the review deals primarily with platelet-HIV interactions. This should be reflected also in the title.

We agree with the reviewer that the title needs to be altered to focus on platelet-HIV interactions. To emphasize the focus of our review topic, the title has been changed to “Platelet and HIV Interactions and Their Contribution to Non-AIDS Comorbidities”

2. The recognition that platelets can inhibit HIV replication, but also contribute to viral persistence is very important, also clinically. This information is well described with many references, but a little bit difficult to digest. Could the authors prepare here a short table/figure to summarize the key facts?

We agree with the reviewer’s comment and an additional illustration would aid the readers’ ability to understand the main points. To address this concern, figure 2 was created to summarize the key facts in an illustrated form.

3. The authors briefly touched the issue of antiplatelets. There are also many other drugs used for inflammatory diseases, which directly or indirectly affect platelets Are there conditions when antiplatelet drugs should be avoided with PLHW?

We thank the reviewer for the comment. There are no specific guidelines on the use of antiplatelet therapy in people living with HIV. The indications and contraindications on the use of antiplatelet agents (such as aspirin, clopidogrel, or prasugrel) in HIV is similar to that of the general population. Antiplatelet agents may be used for primary and secondary prevention of cardiovascular disease, but the risks and benefits must be weighed.  We’ve included this information in section 3.4 (lines 401-403).

4. While this review focusses on platelet HIV-interactions, it might help the reader if the authors have a very short paragraph comparing the HIV interactions with those of other viruses.

The first paragraph in section 2.3 (lines 148-53) was reformatted to include several comparisons between HIV and several other viruses in the context of platelet internalization and platelet binding, resulting in establishment of infection.

Reviewer 2 Report

Comments and Suggestions for Authors

Awamura et al. present a review manuscript on the contribution of platelets in non-AIDS comorbidities.

1. First of all, the title does not reflect the way this manuscript is trying to convey. Almost all of the manuscript discusses the relationship/connection between different conditions with HIV. 

2. Paragraph starting in line 103  - the text flow is poorly structured. Prelude is needed as it is unclear whether HIV activates platelets or authors imply that already activated are more capable of HIV particle endocytosis. 

3. What dose-dependent mean in line 130?

4. In line 132 - again it is not clear what is pre-exisiting, HIV or PLT activation?

5. What do the authors mean by hyperactivation of platelets? This terminology in hemostasis field means the formation of procoagulant platelets. 

6. How do the authors propose HIV sensitizes platelets? This is not discussed at all although the manuscript tries to make it a central aspect. 

7. Can authors define what hyperproliferative thrombocytopenia means in paragraph in line 202?

8. From what the authors presented in paragraph in line 247 there is absolutely no linkage between HIV and HUS. Would it rather be considered a comorbidity?

Minor: 

1. Citation is needed in the sentence in line 45. 

2. Citation is needed for line 91. 

3. Citation is needed for line 92. 

4. Citation 16 in line 151 is misused. 

5. Citation 16 in line 160 is misused. 

6. Citation is needed for line 272. 

7. Citation is needed for line 346. 

8. Citation in line 307 is misused. 

All of these deficiencies have to be addressed before manuscript can be considered for publication. 

Comments on the Quality of English Language

No comments. All is looking good.

Author Response

We would like to thank the Reviewer for their valuable time in reviewing and providing feedback to enhance our manuscript. We have carefully considered the feedback and addressed each concern within the newly submitted manuscript. Below, find included a point-by-point response to each comment.

1. First of all, the title does not reflect the way this manuscript is trying to convey. Almost all of the manuscript discusses the relationship/connection between different conditions with HIV. 

We agree with the reviewer that the title needed alterations. The title was altered to convey the main topics of the paper, in particular platelet-HIV interactions and non-AIDS comorbidities resulting from platelet dysregulation and dysfunction.

2. Paragraph starting in line 103  - the text flow is poorly structured. Prelude is needed as it is unclear whether HIV activates platelets or authors imply that already activated are more capable of HIV particle endocytosis. 

In order to address structuring, a short description of platelet activation in general to address any confusion regarding HIV-platelet activation to serve as a prelude in discussing HIV activation of platelets.

3. What dose-dependent mean in line 130.

According to Tsegaye et. al. activated platelets co-cultured with CD4+ T-cells had lower rates of infection by HIV when the ratio between platelets and CD4+ T-cells was higher. However, as this was in-vitro data, the sentence was altered to reflect this. The term dose-dependent was also omitted to remove and confusion (lines:138-141)

4. In line 132 - again it is not clear what is pre-exisiting, HIV or PLT activation

To address this concern, we have specified that platelets become activated in response to the presence of HIV, which in turn can provide protection against infection in CD4+ T-cells.

5. What do the authors mean by hyperactivation of platelets? This terminology in hemostasis field means the formation of procoagulant platelets. 

We appreciate this insight, as we were unaware that platelet hyperactivation had multiple definitions, making it confusing to the reader. To address this concern, and to also keep consistency with the rest of the manuscript, we have referred to this phenomenon as simply “platelet activation”.

6. How do the authors propose HIV sensitizes platelets? This is not discussed at all although the manuscript tries to make it a central aspect. 

Unfortunately, we were unable to identify what platelet sensitization by HIV is referring to. However, we have expanded the section on platelet activation in general, to make it more clear to the reader of the phenotypic shifts that occur during this process (lines: 103-107). We hope this may clear up any confusion regarding HIV activation of platelets.

7. Can authors define what hyperproliferative thrombocytopenia means in paragraph in line 202?

We have included medical definition (thrombocytopenia due to platelet underproduction; line 217)

8. From what the authors presented in paragraph in line 247 there is absolutely no linkage between HIV and HUS. Would it rather be considered a comorbidity?

TTP/HUS are in the spectrum of thrombotic microangiopathies, are both associated with HIV, typically occur in the later stages of infection, and are both related to immune activation (either antibody-mediated [TTP] or complement mediated [HUS]). We have revised paragraph in line 263-265 to be clearly stated the main points.

Minor comments: (1-8)

Thank you for your catch of this missed detail. We corrected indicated citation accordingly.

Round 2

Reviewer 2 Report

Comments and Suggestions for Authors

The authors addressed all the comments this review had.